# Distinct mechanisms underlie oral vs aboral regeneration in the cnidarian *Hydractinia echinata*

Brian Bradshaw[1], Kerry Thompson[2], Uri Frank[1]*

[1]School of Natural Sciences and Regenerative Medicine Institute, National University of Ireland, Galway, Galway, Ireland; [2]Centre for Microscopy and Imaging, Discipline of Anatomy, School of Medicine, National University of Ireland, Galway, Galway, Ireland

**Abstract** Cnidarians possess remarkable powers of regeneration, but the cellular and molecular mechanisms underlying this capability are unclear. Studying the hydrozoan *Hydractinia echinata* we show that a burst of stem cell proliferation occurs following decapitation, forming a blastema at the oral pole within 24 hr. This process is necessary for head regeneration. Knocking down *Piwi1*, *Vasa*, *Pl10* or *Ncol1* expressed by blastema cells inhibited regeneration but not blastema formation. EdU pulse-chase experiments and in vivo tracking of individual transgenic *Piwi1*$^+$ stem cells showed that the cellular source for blastema formation is migration of stem cells from a remote area. Surprisingly, no blastema developed at the aboral pole after stolon removal. Instead, polyps transformed into stolons and then budded polyps. Hence, distinct mechanisms act to regenerate different body parts in *Hydractinia*. This model, where stem cell behavior can be monitored in vivo at single cell resolution, offers new insights for regenerative biology.

*For correspondence: uri.frank@nuigalway.ie

Competing interests: The authors declare that no competing interests exist.

## Introduction

Cnidarians are renowned for their remarkable ability to regenerate any missing body part. Classical work on the freshwater polyp *Hydra* has shown that both head and foot regeneration can occur without a significant contribution from cell proliferation (i.e., through morphallaxis) (*Park et al., 1970*; *Marcum and Campbell, 1978a*, *1978b*; *Cummings and Bode, 1984*; *Dübel and Schaller, 1990*; *Holstein et al., 1991*). In planarians, by contrast, proliferation of pluripotent stem cells (called neoblasts) and formation of a mass of undifferentiated cells (called blastema) are required for head, tail, and pharynx regeneration (*Reddien and Sanchez Alvarado, 2004*; *Baguñà, 2012*; *Reddien, 2013*; *Adler et al., 2014*). The establishment of a blastema in regeneration has been observed in other taxa including annelid worms (*Bely, 2014*) and echinoderms (*Candia Carnevali, 2006*; *Kondo and Akasaka, 2010*), but the nature of the cells involved is unclear. Urodele amphibians are the only vertebrate tetrapods that can regenerate amputated limbs as adults. They are similar to planarians in their requirement for cell proliferation and blastema formation to complete regeneration, but the cellular source for urodele regeneration is different. In newts, dedifferentiation of cells in the stump provides progenitor cells, but in the axolotl, resident stem cells fulfill the same task (*Sandoval-Guzman et al., 2014*). Furthermore, amphibian blastema cells are lineage restricted rather than being pluripotent (*Kragl et al., 2009*).

The ability to regenerate varies greatly among animals (*Sánchez Alvarado, 2000*; *Sánchez Alvarado and Tsonis, 2006*; *Galliot and Ghila, 2010*; *Tanaka and Reddien, 2011*), with substantial differences sometimes found between closely related taxa: Amphibians, urochordates, planarians, and cnidarians all include both groups or species with excellent regenerative capabilities and their poorly regenerating close relatives (*Sánchez Alvarado, 2000*; *Galliot and Ghila, 2010*). One possible explanation for these observations is that the basic genetic toolkit for regeneration is primitive and present in all animals, but that modulation or loss of some components can modify the ability of a given

**eLife digest** Although all animals are capable of regenerating damaged tissue to some extent, a few—including jellyfish, coral, and their relatives—are able to regenerate entire lost body parts. Closely related species may have very different regeneration capabilities. This has led some researchers to propose that higher animals, such as mammals, still possess the ancient genes that allow entire body parts to regenerate, but that somehow the genes have been disabled during their evolution. Studying animals that can regenerate large parts of their bodies may therefore help scientists understand what prevents others, including humans, from doing so.

An animal that is particularly useful for studies into regeneration is called *Hydractinia echinata*. These tiny marine animals make their homes on the shells of hermit crabs. They are small, transparent and stay fixed to one spot, making it easy for scientists to grow them in the laboratory and closely observe what is going on when they regenerate.

Bradshaw et al. genetically engineered *Hydractinia* individuals to produce a fluorescent protein in their stem cells; these cells have the ability to become one of several kinds of mature cell, and often help to repair and grow tissues. This allowed the stem cells to be tracked using a microscope.

When the head of *Hydractinia* was cut off, stem cells in the animals' mid body section migrated to the end where the head used to be and multiplied. These stem cells then created a bud (known as a blastema) that developed into a new, fully functional head within two days, allowing the animals to capture prey. Reducing the activity of certain stem cell genes prevented the new head from growing, but the bud still formed.

Next, Bradshaw et al. removed a structure from the opposite end of the animal, called the stolon, which normally helps *Hydractinia* attach to hermit crabs shells. Stolons regenerated in a completely different way to heads. No bud formed. Instead, the remainder of the animal's body, which included the head and the body column, gradually transformed into a stolon rather than regenerating this structure, and only then grew a new body column and head. Therefore, different tissues in the same animal can regenerate in different ways. Understanding the 'tricks' used by animals like *Hydractinia* to regenerate may help translate these abilities to regenerative medicine.

taxon to regenerate. This has recently been shown to be the case in planarians where changes in canonical Wnt signaling underlie differences in regenerative ability between closely related species (*Liu et al., 2013*; *Sikes and Newmark, 2013*; *Umesono et al., 2013*). Hence, studying regeneration in a broad variety of animal models might reveal both regeneration mechanisms that are primitive and widely shared among animals as well as evolutionarily derived ones and could assist in addressing a major question in regenerative medicine, namely why humans are not capable of regenerating many tissues.

A specific difficulty in the study of tissue and organ renewal in higher animals is the fact that, like embryonic development, regeneration is a dynamic process. Therefore, understanding regeneration requires the analysis of individual cells over long time periods covering the duration of the regenerative process. The large size and opaque nature of many animals impede in vivo regeneration research at such resolution in most model organisms.

In this study, we show that *Hydractinia echinata,* which is a common colony-forming cnidarian in the European North Atlantic (*Figure 1*), provides a powerful model system to study the cellular and molecular basis of animal regeneration. Indeed, *Hydractinia* is easy to culture in the lab and allows whole mount gene expression analysis, cellular analyses, transgenesis, and gene knockdown (*Plickert et al., 2012*). The animal reproduces sexually on a daily basis, but also grows clonally by elongation of gastrovascular tubes, called stolons, and asexual budding of new polyps (*Figure 1*). Finally, they are small and optically translucent, and post-metamorphic animals are sessile and can grow on microscope slides enabling in vivo imaging of cellular processes. Like many cnidarians, *Hydractinia* displays a remarkable regenerative ability and growth plasticity, but the molecular and cellular mechanisms underlying these capabilities are not well understood. We have studied both oral (i.e., head) and aboral (i.e., stolon) regeneration in *Hydractinia* and show that stem cell migration and proliferation underlie head regeneration in this animal. Surprisingly, stolon regeneration is achieved through a fundamentally different cellular and morphogenetic process, demonstrating that a single species can apply different mechanisms to regenerate different tissues or body parts.

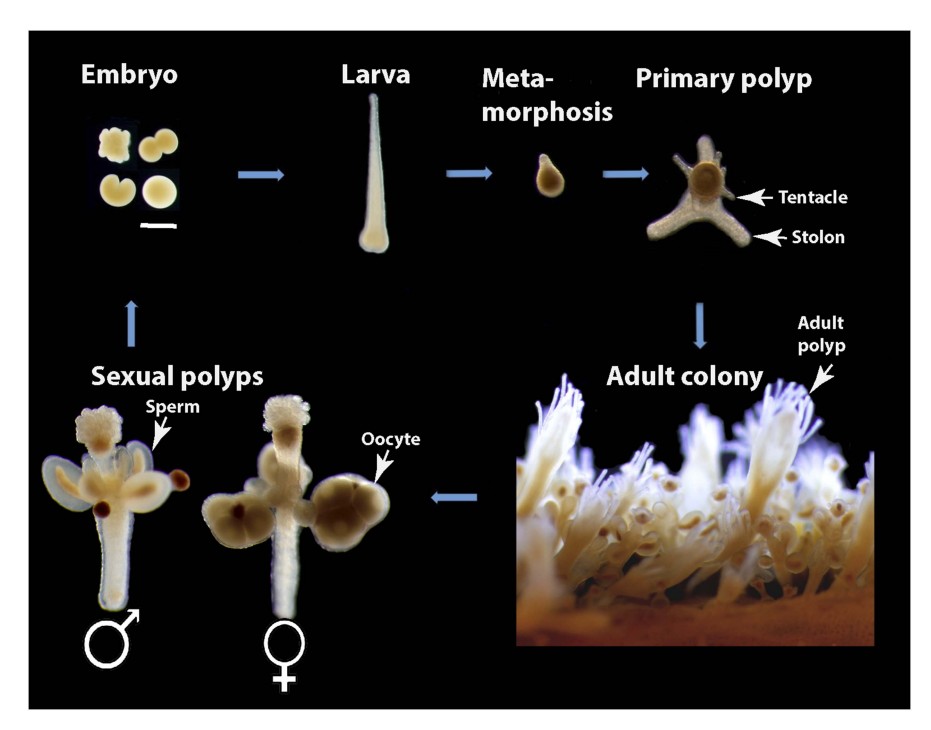

**Figure 1**. *Hydractinia* life cycle and colony structure. Scale bar 200 μm.

## Results

### Head regeneration in *Hydractinia*

Our first aim was to characterize *Hydractinia* head regeneration. For this, polyps were isolated from their colony by a transverse cut close to the polyp-stolon boundary and decapitated (n > 300; *Figure 2A*). The remaining cylinder-like body column was then followed until a new head developed (*Figure 2B*) and the animals regained the ability to feed. Lesion closure by stretching out of epidermal epithelial cells occurred at both cut ends within 6 hr following decapitation in all cases. No further development occurred at the aboral end where the stolons were removed (see section below). About 24 hr later, a dome-like tissue appeared at the oral pole. This was followed by the development of a new mouth and tentacles 48–96 hr post decapitation (in over 95% of cases; *Figure 2B*). Anti-acetylated tubulin antibody staining confirmed that the nervous system had completely regenerated with both neurons and nematocytes (cnidarian-specific mechanosensory/effector cells) appearing normal as in control animals (*Figure 2C–F*). The polyps regained the ability to catch prey about two to 3 days post decapitation when a new head fully formed, but tentacle elongation sometimes continued for an additional few days. The time course of head regeneration (n > 300) was variable among polyps, lasting between 1 and 4 days, depending on age (young post metamorphosis animals may regenerate faster), genetic background (we use a polymorphic wild type laboratory population), and general state of health (malnourished or otherwise unhealthy animals may display delayed regeneration). No indications for stolon regeneration were observed within the time course of head regeneration. Regeneration of decapitated polyps that remained attached to their colonies was indistinguishable from isolated polyps within the natural variability stated above.

### Cell proliferation during regeneration

Head regeneration after decapitation in the freshwater cnidarian, *Hydra*, can occur in the absence of cell proliferation (*Park et al., 1970*; *Marcum and Campbell, 1978a*, *1978b*; *Dübel and Schaller, 1990*; *Holstein et al., 1991*). We therefore decided to analyze cell proliferation in *Hydractinia* head

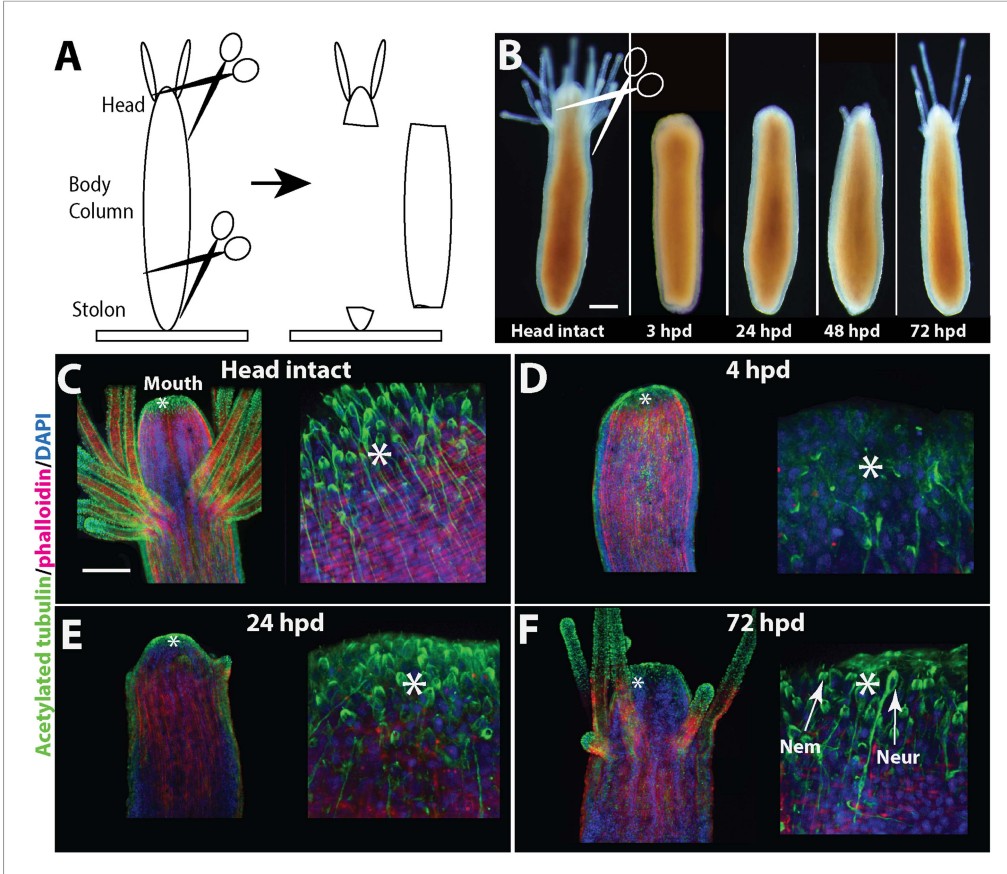

**Figure 2**. Head regeneration in *Hydractinia*. (**A**) Experimental setup. (**B**) Live images of regenerating polyp. Scale bar 100 µm. (**C**–**F**) Anti acetylated tubulin (green)—phalloidin (red)—DAPI (blue) staining of regenerating polyp. Asterisks are depicted at approximately the same position in each panel. (**C**) Intact polyp. (**D**) 4 hpd. (**E**) 24 hpd. (**F**) 72 hpd. Nem = nematocyte; Neur = neuron. Scale bars 100 µm.

regeneration. In intact *Hydractinia* polyps, cell proliferation was almost exclusively restricted to a band at the lower part of the polyp body column, with little or no proliferating cells outside of this band. This was shown with both EdU incorporation as well as anti-phospho-Histone 3 (pH3) antibody labeling to visualize mitotic cells (*Figure 3A*). We then decapitated polyps and allowed them to regenerate. During the course of regeneration, we incubated the polyps in EdU for 40 min at different times following decapitation. Animals were immediately fixed and processed for EdU visualization or anti pH3 staining. These experiments showed, first, that wound closure was not associated with enhanced cell proliferation (*Figure 3—figure supplement 1*). However, a striking shift in the spatial distribution of cycling cells was evident 24–48 hpd (hours post-decapitation). In contrast to the lower body column band-fashion distribution of cycling cells in intact polyps, the post decapitation distribution of cycling cells concentrated at the oral pole where the new head was about to form (n = 100; *Figure 3A*). Intact polyps that were labeled by EdU while still connected to their stolonal network showed the same pattern of S-phase cell distribution as polyps with intact heads isolated from their colonies (n = 20; *Figure 3—figure supplement 1*). Hence, head but not stolon amputation in *Hydractinia* is followed by the formation of a blastema with highly proliferative cells.

## Proliferative cells during regeneration

We then set to address the question of which cell types proliferated during regeneration. For this, we decapitated animals and 24 hr later macerated them as previously described (*David, 1973*). We also macerated intact animals as control. Cells were spread on glass slides and stained with anti pH3

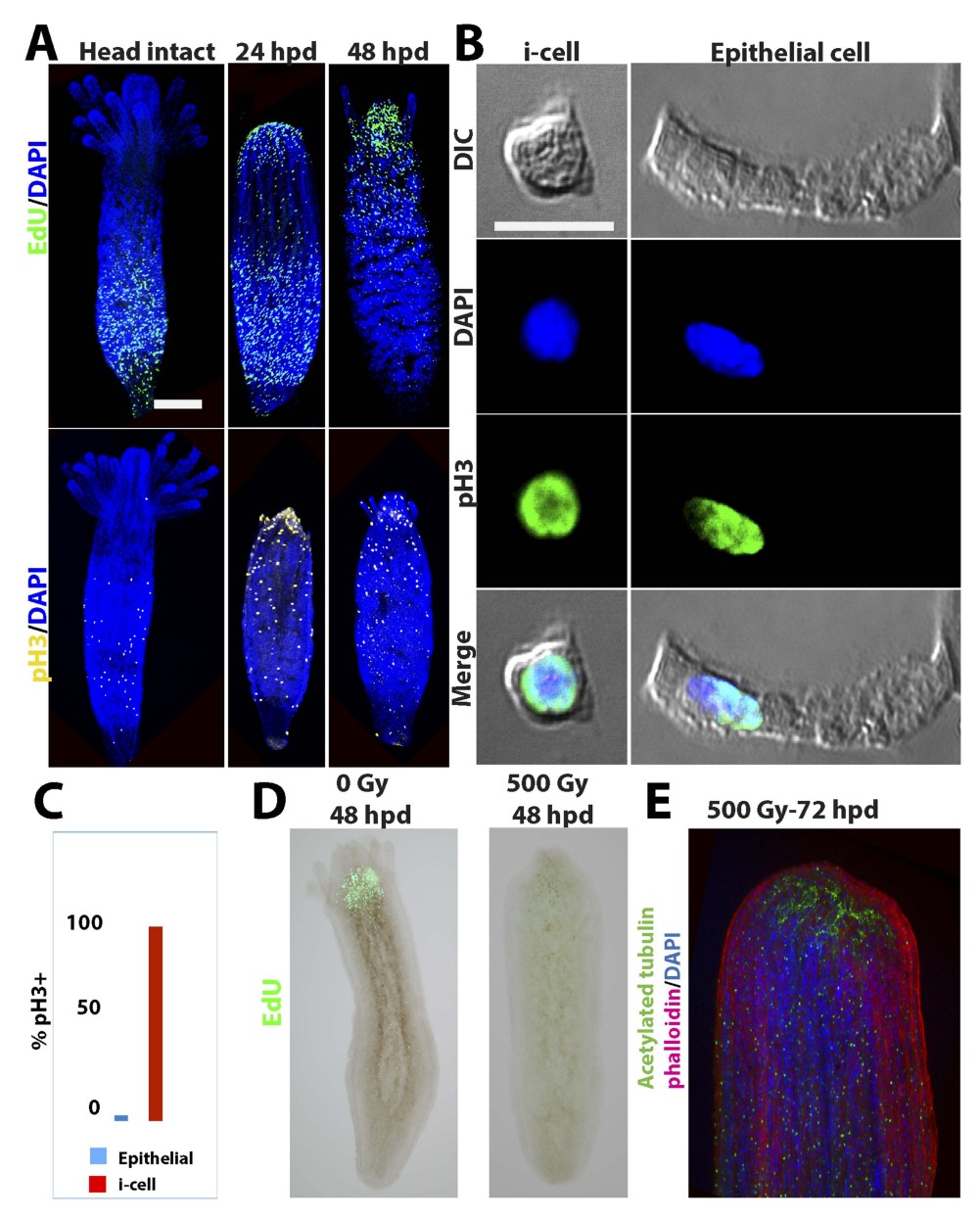

**Figure 3**. Cell proliferation during head regeneration. (**A**) EdU (upper panes) and phospho-Histone 3 (pH3) (lower panes) labeling of cells. Scale bar 200 µm. (**B**) Maceration of animals and labeling of cycling cells. Scale bar 10 µm. (**C**) Percentages of epithelial and i-cells out of the total mitotic cells. (**D**) Effect of gamma irradiation on cell proliferation and regeneration. (**E**) Effect of irradiation on nervous system regeneration. Green, anti-acetylated tubulin; red, phalloidin; blue, DAPI.

The following figure supplements are available for figure 3:

**Figure supplement 1**. EdU labeling of polyps.

**Figure supplement 2**. A selection of dissociated Hydractinia cells.

**Figure supplement 3**. Effect of different absorbed doses of gamma irradiation on head regeneration and cell proliferation.

*Figure 3. continued on next page*

*Figure 3. Continued*

**Figure supplement 4**. TUNEL staining of control and irradiated polyps post decapitation.

**Figure supplement 5**. Effect of 30 µM mitomycin C on cell proliferation and head regeneration.

antibodies. Cnidarians have relatively few cell types including epidermal and gastrodermal myoepithelial cells, several types of neurons, stinging cells (nematocytes), gland cells, and stem cells (*Figure 3—figure supplement 2*). Hydrozoan stem cells are called interstitial cells, or i-cells for short. In *Hydractinia*, i-cells reside in interstitial spaces between epithelial cells (mostly epidermal), and at a population level are thought to be pluripotent life long, giving rise to all somatic lineages as well as germ cells (*Müller et al., 2004*). This is very different from *Hydra* where i-cells do not contribute to the two self-renewing epithelial lineages (*Bode, 1996*). Most cell types are easily distinguishable morphologically (*Figure 3—figure supplement 2, 3B*). We have counted mitotic cells in intact vs regenerating animals 24 hpd in three separate experiments and found that on average epithelial cells, as identified by morphology, form less than 4% of the total mitotic cell complement, whereas the vast majority of S-phase cells are i-cells (*Figure 3B,C*). There was no significant difference in the relative proportion of mitotic epithelial cells in intact vs regenerating animals. Hence, i-cells form the major proliferative cellular component in *Hydractinia* head regeneration, but the contribution of epithelial cells to this process through proliferation and/or transdifferentiation cannot be ruled out based on these data.

## Cell proliferation is required for regeneration

To address the requirement for cell proliferation for head regeneration we exposed *Hydractinia* polyps to gamma irradiation. We first established that absorbed doses of up to 300 Gy did not completely abolish cell proliferation in decapitated polyps (n = 30; *Figure 3—figure supplement 3*). At 500 Gy, S-phase cells were no longer detectable (n = 30; *Figure 3—figure supplement 3*) but the animals remained responsive to mechanical stimulation.

To analyze the direct effect of gamma irradiation and assess cell death by apoptosis we performed TUNEL staining on intact and decapitated animals that had or had not been irradiated. We found small numbers of apoptotic cells in non-irradiated animals between 6 and 24 hpd (n = 30). This is consistent with previous work on other animals, where apoptotic cells were shown to play a role in the regenerative process (*Hwang et al., 2004*; *Tseng et al., 2007*; *Chera et al., 2009*; *DuBuc et al., 2014*). In irradiated animals, the distribution of apoptotic cells was similar to the distribution of proliferating i-cells in non-irradiated animals (n = 30; *Figure 3—figure supplement 4*; compare with *Figure 3A*). We concluded that cycling i-cells are sensitive to gamma irradiation.

We then irradiated animals at 500 Gy followed by decapitation, and returned them to their culture tanks. None of the animals regenerated at 48 hr following this treatment (n = 30; *Figure 3D,E*; *Figure 3—figure supplement 3*). Animals irradiated at 300 Gy or lower did regenerate, but at a markedly slower pace (n = 30; *Figure 3—figure supplement 3*). Importantly, the lack of EdU incorporation showed that no proliferative blastema developed at the oral pole of animals irradiated at 500 Gy (*Figure 3D*). Treatment with mitomycin C, a cytostatic drug that was shown to kill i-cells in *Hydractinia* (*Müller et al., 2004*), had similar effects, with animals failing to regenerate following treatment (*Figure 3—figure supplement 5*). Therefore, cell proliferation and blastema formation are essential for *Hydractinia* head regeneration. This is markedly different from *Hydra* head regeneration, which can occur through morphallaxis in the complete absence of cycling cells.

## Genes acting during head regeneration

Our next step was to study gene expression during regeneration. We focused on *Piwi1*, *Vasa*, *Myc2* and *Pl10*, all standard stem cell markers in cnidarians and other metazoans (*Reddien et al., 2005*; *Rebscher et al., 2008*; *Voskoboynik et al., 2008*; *Alie et al., 2011*; *Collins et al., 2013*; *Juliano et al., 2014*). In addition, we studied *Ncol1*, an early nematocyte differentiation marker in cnidarians (*David et al., 2008*; *Millane et al., 2011*). We first established the normal expression pattern in intact polyps using in situ hybridization. As shown for *Vasa* previously (*Rebscher et al., 2008*), all i-cell

marker genes were expressed in a band fashion at the lower part of the polyp body column, co-localizing with S-phase and mitotic cells (*Figure 4A,B*; compare with *Figure 3A*; n = 40).

We then decapitated polyps and fixed them at different time points during head regeneration and performed in situ hybridization with cRNA probes for i-cell genes. We found that decapitation had a major effect on the distribution of i-cells. Instead of being restricted to the band area, we now saw i-cells at more oral positions. Most strikingly, about 24 hpd, coinciding with the major proliferative peak in the blastema, there was strong expression of i-cell markers and *Ncol1* in the blastema (n = 40; *Figure 4C,D*). Hence, the blastema that forms during head regeneration at the oral pole contains large numbers of i-cells and early nematoblasts in contrast to the absence of these cells in oral areas in intact animals.

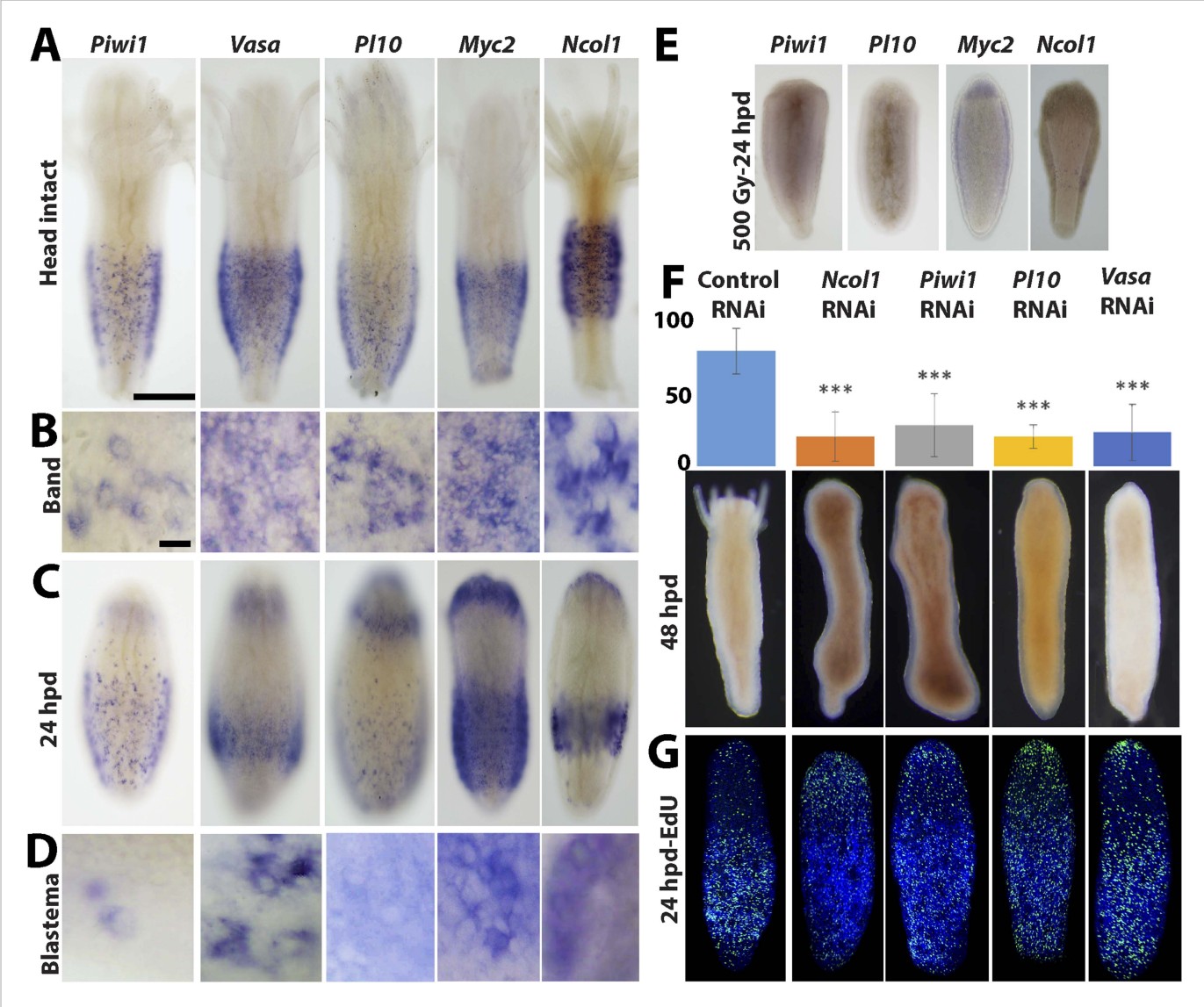

**Figure 4**. Gene expression during head regeneration. (**A**) Expression of i-cell marker genes in intact polyps. Scale bar 200 μm. (**B**) Higher magnification of positive cells in the band. Scale bar 10 μm. (**C**) Expression of marker genes 24 hr post decapitation. (**D**) Higher magnification of blastema cells expressing i-cell marker genes. (**E**) Effect of gamma irradiation on i-cell marker gene expression. (**F** and **G**) Downregulation of marker genes during regeneration by RNAi. (**F**) Effect on regeneration and quantitative analysis of knockdown. (**G**) Effect on cell proliferation.

The following figure supplement is available for figure 4:

**Figure supplement 1**. Effect of histones H2A and H4 knockdown and quantification of *Piwi1* mRNA following RNAi knockdown.

Irradiation of animals at 500 Gy resulted in a marked reduction of i-cells and nematoblast marker expression at 24 hr post decapitation (n = 10; *Figure 4E*). This further shows that i-cells are sensitive to irradiation and are required for head regeneration.

## Downregulation of i-cell genes inhibits regeneration

To study the role of stem cell genes in head regeneration we used RNAi to knockdown *Piwi1, Vasa, Pl10* and *Ncol1*. RNAi was performed as previously described (*Duffy et al., 2010, 2011; Millane et al., 2011*). Briefly, polyps were removed from their colony and were then decapitated. Following decapitation the cylindrical body columns were incubated in seawater containing 30–50 ng/µl double stranded RNA (dsRNA) corresponding to 200 bp coding sequence of *Piwi1, Vasa, Pl10* and *Ncol1*. Control experiments were performed with dsRNA corresponding to the backbone sequence of the pBlueScript cloning vector, which is not encoded by the *Hydractinia* genome. qPCR analysis of *Piwi1* RNAi treated regenerating animals revealed a significant reduction in *Piwi1* mRNA levels comparing to control RNAi (*Figure 4—figure supplement 1*).

Regeneration in animals in which *Piwi1, Vasa, Pl10* or *Ncol1* were knocked down was compromised, with the frequency of regenerating RNAi animals significantly different from the control RNAi (chi-square test, p = 0.00001; *Figure 4F*). Animals treated with control (i.e., non-coding) dsRNA regenerated normally. The experiments were repeated four times with 10 animals for each treatment. To address the role of these genes in i-cell proliferation, we treated regenerating animals with dsRNA as described above. Twenty-four hours after decapitation we incubated them in EdU for 20 min followed by fixation and EdU visualization. We found that knockdown of *Piwi1, Vasa, Pl10,* or *Ncol1* had no visible effect on cell proliferation (*Figure 4G*). In these RNAi animals the head blastema formed normally yet regeneration was significantly affected, suggesting that these genes are required for differentiation. The specificity of the RNAi treatment was further confirmed by knockdown of histones H2A or H4 which strongly reduced EdU incorporation in regenerating animals (*Figure 4—figure supplement 1*). Hence, *Piwi1, Pl10,* and *Vasa* expression is not required for i-cell proliferation and for the formation of the head blastema. *Ncol1*, which is a nematogenesis marker, is not expressed in cycling i-cells and its downregulation is therefore not expected to affect S-phase cells.

## The cellular source of head blastema is migration of stem cells

The experiments described above show that decapitation results in head blastema formation that includes numerous proliferating i-cells. In fully grown, intact animals, i-cells are more or less restricted to the band area in the lower polyp body column, and are nearly absent from the head, which also does not include significant numbers of proliferating cells (n = 75/83). We therefore reasoned that in the near absence of resident stem cells in the intact head (*Figure 4A*), the source of stem cells in the head blastema could either be migration, that is, i-cells moving from the band in the lower polyp body column, or dedifferentiation of local differentiated cells in the stump. To discriminate between these two options, we performed two types of experiments. First, we pulse-labeled intact polyps with EdU for 60 min, followed by decapitation. The animals were intensively washed to remove EdU and left to initiate head regeneration. They were fixed and processed for EdU visualization at 6 and 24 hpd. We found that animals fixed 6 hpd had a strong signal in the band at the lower body column and only a few EdU⁺ cells were scattered at more oral positions (*Figure 5A*). However, those animals fixed 24 hpd had also a strong EdU signal in the developed blastema (*Figure 5B; Figure 5—figure supplement 1*). Because no EdU could be incorporated after washing the animals, the EdU⁺ cells in the blastema must have been the very same cells that were in S-phase in the lower band 24 hr earlier and had migrated to the stump following decapitation. Analysis of pH3 immunoreactivity in EdU pulse chased animals revealed double positive cells in the blastema, indicating that S-phase cells in the band continued to proliferate even after reaching the stump and contributed new cells to the blastema through mitosis (*Figure 5—figure supplement 1*).

Second, to gain a more dynamic view on i-cell migration during regeneration we established transgenic animals expressing GFP under the *Hydractinia* endogenous *Piwi1* genomic control elements. For this, we cloned 2.5 kb upstream and 1.1 kb downstream of the *Piwi1* genomic coding sequence locus and inserted the *GFP* coding sequence instead of *Piwi1* (*Figure 5—figure supplement 2*). This construct was microinjected to one-cell stage embryos as described previously (*Künzel et al., 2010*). Genomic integration and stable GFP expression occurs in *Hydractinia* within 24 hr (*Künzel et al., 2010*). GFP expression in transgenic polyps (*Figure 5C,D*) was consistent with the

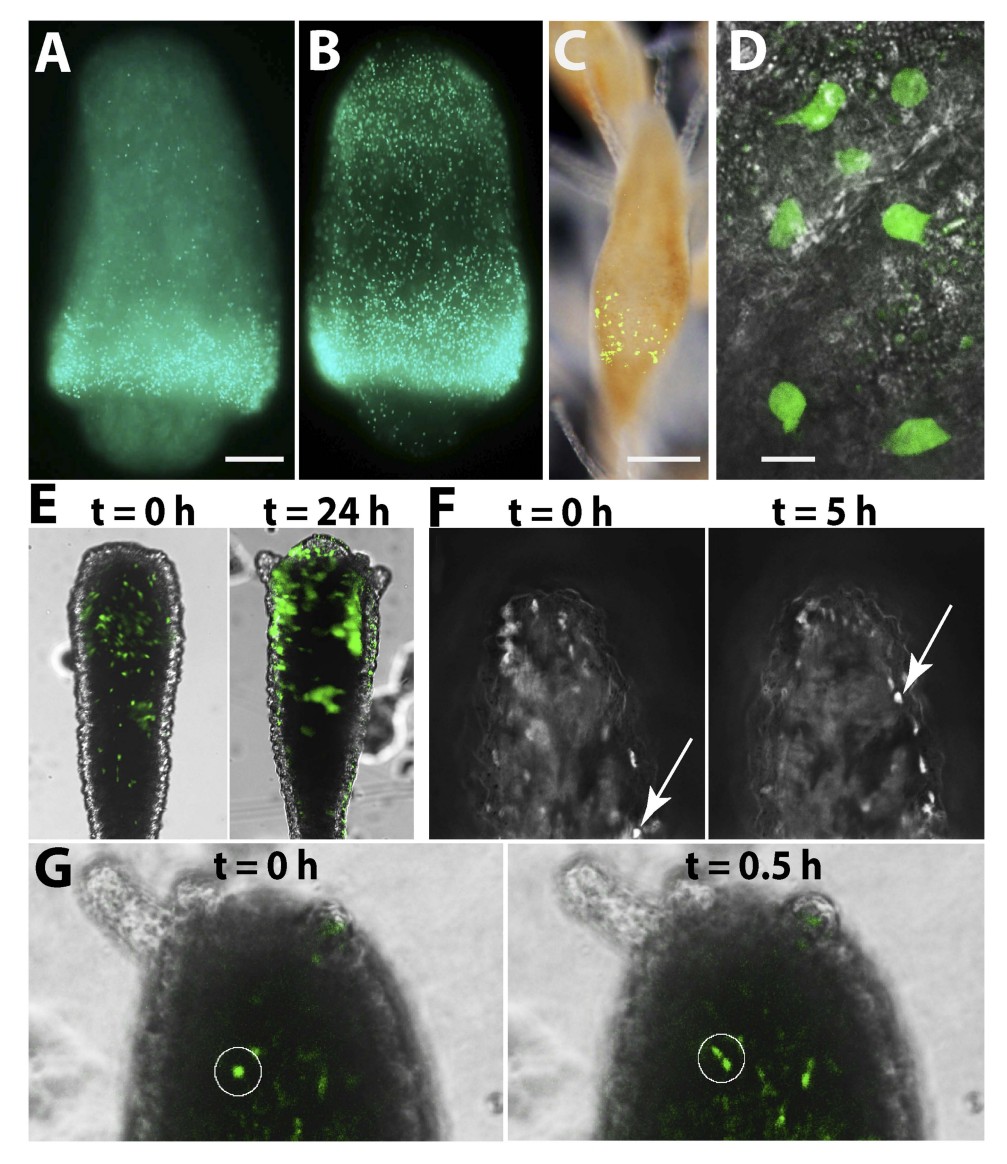

**Figure 5**. The cellular source for head regeneration. (**A** and **B**) EdU pulse-chase. (**A**) 6 hpd—most EdU⁺ cells restricted to the band. Scale bar 50 μm. (**B**) 24 hpd—many EdU⁺ cells migrated to the blastema. (**C–G**) Live images of *Piwi1*-GFP⁺ transgenic cells. (**C**) Transgenic *Piwi1*-GFP⁺ polyp. Scale bar 200 μm. (**D**) Higher magnification of live GFP⁺ i-cells in vivo. Scale bar 10 μm. (**E**) Live transgenic polyp pictured at 10 (left) and 24 (right) hpd. (**F**) Live images of a single, *Piwi1*-GFP⁺ i-cell migrating to the forming blastema (arrow). (**G**) Live image of a single, *Piwi1*-GFP⁺ i-cell dividing during migration (encircled).

The following figure supplements are available for figure 5:

**Figure supplement 1**. Proliferation of migrating cells.

**Figure supplement 2**. The structure of the construct used to generate transgenic, *Piwi1*-GFP+ animals.

**Figure supplement 3**. Piwi1 immunohistochemistry and co-localization of gene expression and S-phase cells.

---

endogenous expression of *Piwi1* as assessed by in situ hybridization (*Figure 4A*) and immunohisto-chemistry (*Figure 5—figure supplement 3*). Because genomic integration does not occur in all cells, the animals were mosaics and not all *Piwi1*⁺ i-cells expressed GFP. This feature was useful because the

density of GFP+ cells was not as high as would be expected in a fully transgenic animal, facilitating tracking of single cells in vivo (*Figure 5C,D*).

Transgenic polyps were isolated from their colonies by a transverse cut close to the polyp-stolon boundary. Polyps were decapitated and then viewed while the head blastema was developing over several hours using a fluorescence stereomicroscope (*Figure 5C*), time-lapse DeltaVision deconvolution microscope (*Figure 5F*; *Video 1*), or Andor spinning disk confocal microscope (*Figure 5G*; *Video 2*). *Piwi1+* cells were observed migrating into the prospective head area (*Figure 5E–G*; *Videos 1, 2*) and no evidence for dedifferentiation was evident as all viewed recruited GFP+ cells at the blastema were migratory. Some migrating cells underwent mitosis before reaching the blastema; they stopped migrating, completed mitosis, and the two daughter cells resumed migration (*Figure 5G*; *Video 2*). Based on these experiments and the EdU pulse-chasing, we conclude that the primary cellular source for establishing the head blastema and, subsequently, head regeneration is migration of i-cells from the band in the polyp lower body column to the prospective head area. A possible contribution of existing epithelial cells to the regeneration process through mitosis and/or transdifferentiation (body column epithelial cells to tentacle epithelial cells) cannot be ruled out.

## Stolon regeneration involves different mechanisms than head regeneration

*Hydractinia* polyps are not able to regenerate stolons directly from their aboral ends (*Müller et al., 1986*). Polyps, or even fragments of them, can, instead, transform into stolons but this phenomenon is not well understood (*Putnam Hazen, 1902*; *Müller et al., 1986*; *Lange and Müller, 1991*). To study aboral, that is, stolon, regeneration we removed polyps from their colony by a transverse cut at the lower third of the polyp body column to exclude any stolonal tissue. Isolated polyps healed the cut surface within hours (*Figure 6A*). We then followed the polyps and photographed them every 24 hr for up to 25 weeks. The polyps appeared normal for days and sometimes weeks, responded to mechanical stimuli by contraction, and were able to catch, kill, and ingest brine shrimp nauplii, resembling a solitary *Hydra* polyp. No blastema developed at the aboral pole after stolon removal, and the general distribution of EdU+ cells was largely similar to polyps that were labeled on their colony. (*Figure 3—figure supplement 1*). Over the next weeks, however, the polyps started resorbing their tentacles and thereby lost the ability to feed (*Figure 6A*). Next, the entire head structure disappeared and each polyp's cylindrical body column started to elongate, thereby decreasing its diameter (*Figure 6A*). New branches appeared at irregular intervals and some developed into polyps with fully functional heads and tentacles, thereby regaining the ability to feed (n = 200; *Figure 6B*). Chitin secretion (which is stolon specific in *Hydractinia*) (*Lange and Müller, 1991*) commenced (*Figure 6C*) and eventually, sexual polyps developed and produced fertile gametes (*Figure 6D*). Based on chitin secretion and ability to generate polyps, it appeared that the polyp body column had transformed into stolons rather than regenerated new stolons from the aboral stump.

To characterize the molecular events associated with the transformation of polyps to stolons we first studied the expression of *Wnt3* during this process. *Wnt3* is an established oral marker in *Hydractinia* (*Plickert et al., 2006*; *Müller et al., 2007*; *Duffy et al., 2010*), but is also expressed weakly in the i-cell band of polyps (*Plickert et al., 2006*). We found that polyp heads that were losing their tentacles had also

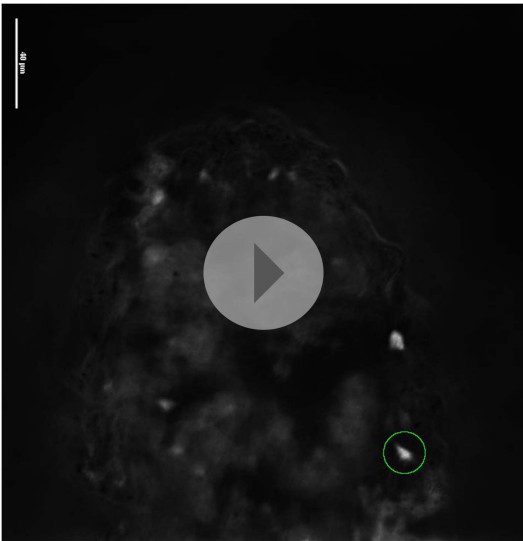

**Video 1.** Follow up of individual cells migrating to forming blastema.

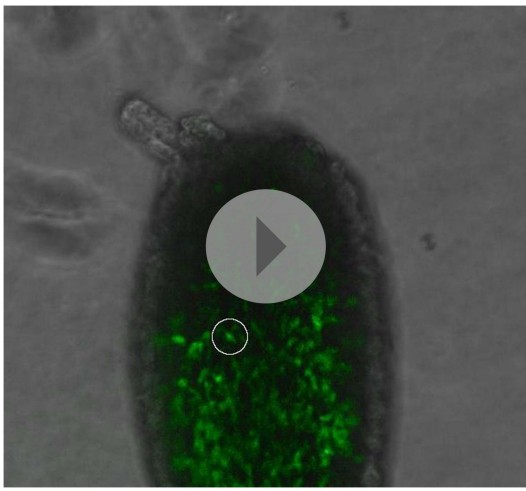

**Video 2.** Follow up on proliferating cell migrating to blastema.

lost *Wnt3* expression (*Figure 6E*). By contrast, *Wnt3* mRNA reappeared in the oral tip of new polyps budding from this newly transformed tissue (*Figure 6F*). The transformed polyp expressed the gene in a more ubiquitous fashion (see below). Hence, loss of oral *Wnt3* expression preceded, or accompanied, the loss of head characteristics such as tentacles and mouth.

To further analyze the polarity of polyps that transformed to stolons we performed in situ hybridization using i-cell marker cRNA probes. In normal, non-regenerating polyps, i-cells were largely restricted to the band area at the polyp lower body column (*Figure 4A*). In stolons, by contrast, i-cells were equally distributed in the epidermal interstices along the stolon flanks (*Figure 6—figure supplement 1*). Hence, we hypothesized that a polyp to stolon transformation should be reflected by a spatial change in i-cell marker expression from restricted band-like to ubiquitous. Indeed, in situ hybridization of *Piwi1, Vasa, Pl10, Myc2* and *Ncol1* on isolated polyps that lost head structures showed a stolon-like expression pattern of these genes (*Figure 6G*; *Figure 6—figure supplement 2*).

To summarize this point, *Hydractinia* aboral regeneration is not direct and proceeds through three stages: First, loss of anterior-posterior polarity; second, full transformation of the polyp into a stolon; third, budding new polyps and regaining oral-aboral polarity. Hence, *Hydractinia* polyps can regenerate a head through i-cell migration and blastema formation, but they cannot directly regenerate stolons; they can transform into stolonal tissue instead.

## Discussion

We have studied both head and stolon regeneration in the cnidarian *Hydractinia.* Decapitation was followed by rapid wound healing that primarily involved stretching of epithelial cells without the requirement for cell proliferation (*Figure 3—figure supplement 1*). Thereafter, we monitored the migration of stem cells (i-cells) from their normal position in the band area at the lower polyp body column to the prospective head, and their proliferation to form a head blastema. Of note, our data show that not all i-cells migrate to the stump, consistent with migratory vs non-migratory i-cell sub-populations. New head structures developed within 2–3 days, after which most i-cells disappeared from the head area and resumed their normal position in the band. Gamma irradiation abolished blastema formation and regeneration altogether. By contrast, individual knockdown of each one of the i-cell genes *Piwi1, Vasa* and *Pl10,* and the early nematogenesis marker *Ncol1* did not prevent blastema formation, but did inhibit regeneration. Hence, the role of these genes might be related to the ability of i-cells to differentiate rather than to keeping them undifferentiated. Similar results were obtained with *Smedwi2* (a *Piwi* homologue) knockdown in planarians (*Reddien et al., 2005*), but knocking out a different stem cell gene, *Sox2*, in axolotl inhibits proliferation of neural progenitors (*Fei et al., 2014*).

A common view in the literature has been that cnidarians can regenerate through morphallaxis, that is, without contribution from cell proliferation (*Park et al., 1970*; *Marcum and Campbell, 1978a*, *1978b*; *Holstein et al., 1991*). More recent studies conducted on *Hydra* and on the sea anemone *Nematostella vectensis*, however, have shown that cell proliferation accompanies the regeneration of cnidarian heads under normal circumstances (*Govindasamy et al., 2014*), but the necessity of proliferation was only demonstrated in *Nematostella* decapitation and *Hydra* bisection (*Miljkovic-Licina et al., 2007*; *Chera et al., 2009*; *Passamaneck and Martindale, 2012*; *DuBuc et al., 2014*). Our results support the new emerging view on head regeneration in the Cnidaria and are consistent with Passamaneck and Martindale's hypothesis (*Passamaneck and Martindale, 2012*) that *Hydra's* ability to regenerate a head in the absence of cell proliferation is evolutionarily

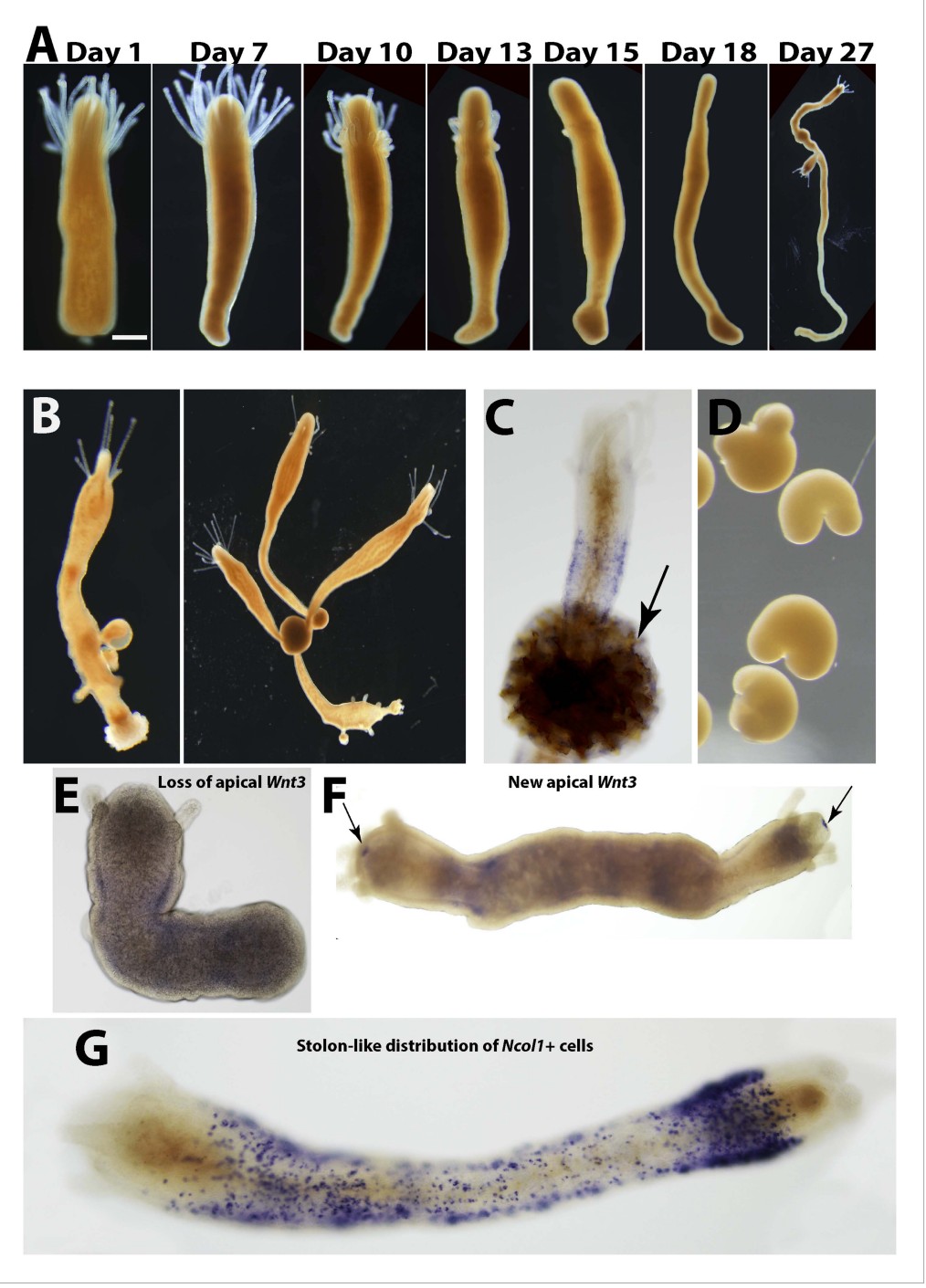

**Figure 6**. Stolon regeneration. (**A**) Time course of a single polyp transforming into a stolon and budding new polyps. Scale bar 200 µm. (**B**) Sexually mature colonies derived from an isolated polyp. (**C**) Chitin secretion (arrow) by a polyp that has transformed into a stolon. (**D**) Early embryos spawned by colony derived from a single polyp. (**E**) Loss of oral *Wnt3* in polyp transforming into a stolon. (**F**) Oral *Wnt3* expression in newly bud polyps (arrows). (**G**) Stolon-like expression of i-cell markers in transformed polyps.

The following figure supplements are available for figure 6:

**Figure supplement 1**. Expression of i-cell markers in stolons and *Wnt3* in a primary polyp.

**Figure supplement 2**. Expression of i-cell and nematogenesis markers in polyps that had transformed into stolons.

derived within this phylum. This scenario suggests that blastema formation is an evolutionarily primitive hallmark of distal regeneration in animals.

Isolated polyps were unable to directly regenerate stolons from their aboral end like they do following decapitation from the oral end, consistent with previous studies (*Putnam Hazen, 1902*; *Müller et al., 1986*; *Duffy et al., 2010*), and no blastema formed at the aboral stump following removal of the stolons (*Figure 3—figure supplement 1*). Instead of regenerating stolons, isolated polyps lost oral-aboral polarity, and polyp identity altogether, and transformed into stolons. This process lasted for many weeks, thereby demonstrating their remarkable growth plasticity. Polyp to stolon transformation was preceded by loss of oral *Wnt3* expression and oral-aboral polarity, and acquisition of a ubiquitous, stolon-like distribution of i-cells, as opposed to the band like distribution typical of polyps. The newly transformed stolons budded new polyps and became fully functional, sexually mature colonies. These data, therefore, show that *Hydractinia* polyps possess tissue pluripotency. For now, however, we cannot discriminate between the scenarios of pluripotent i-cells vs several self-renewing, but lineage restricted, progenitors. So why do polyps not directly regenerate stolons? We suggest that tissue polarity along the oral-aboral axis prevents direct stolon regeneration. In a previous study, it has been shown that Wnt signaling promotes oral structures in *Hydractinia*, but represses stolons (*Duffy et al., 2010*). Downregulation of *Wnt3* or *Tcf* in decapitated polyps induces phenotypes reminiscent of polyp to stolon transformation in the present study, but requires shorter time to develop (*Duffy et al., 2010*). We show that in the absence of experimental manipulation, *Wnt3* expression and oral-aboral polarity are lost spontaneously in isolated polyps, enabling the transformation of the polyp into stolonal tissue that can bud new polyps. Possibly, *Wnt3* not only maintains oral- but also polyp- identity, and stolons develop by default in its absence. A summary of the two distinct regenerative processes in *Hydractinia* is schematically illustrated on *Figure 7*.

In conclusion, our results, and results published over the past few years by others (*Chera et al., 2009*; *Kragl et al., 2009*; *DuBuc et al., 2014*; *Sandoval-Guzman et al., 2014*), show that the mechanisms governing animal regeneration can be not only species-specific, but also tissue-specific within a single species. Some regeneration mechanisms, like blastema formation, are conserved in animals, and their modulation over evolutionary times may have affected the regenerative ability of different species. An exciting development in the study of regeneration is provided by the ability to track individual, transgenic cells in vivo using *Hydractinia* as an animal model. In vivo cell migration assays have been performed in *Hydra* (*Khalturin et al., 2007*), but the sessile nature of adult *Hydractinia* facilitates long-term studies at single cell resolution.

## Materials and methods

### Animal culture
Colonies of *H. echinata* were cultured in artificial seawater at 18°C under 14/10 light/dark regime. They were fed brine shrimp nauplii four times a week and ground fish once a week.

### Animal maceration
Animals were anesthetized for 30 min in 4% MgCl in seawater. They were then placed in a Glycerin/Acetic acid/Seawater (1:1:13) solution for 10 min, followed by incubation in Glycerin/Acetic acid/Distilled Water (1:1:13) for 2 hr. They were then pipetted up and down to complete maceration and fixed in 8% formaldehyde for 30 min. Cells were spread on a glass slide and dried overnight. Cell counting of macerated cells was performed by eye using a FV1000 Olympus confocal scanning laser microscope.

### EdU labeling
EdU incorporation was performed for 20–60 min at a concentration of 150 µM. For visualization, animals were fixed and processed using the Click-iT EdU Alexa Fluor 488 Imaging Kit (Life Technologies, Dun Laoghaire, Co Dublin, Ireland) according to the manufacturer's protocol.

### Gamma irradiation
Gamma-irradiation was carried out using a $^{137}$Cs source at a dose-rate of 12 Gy/min.

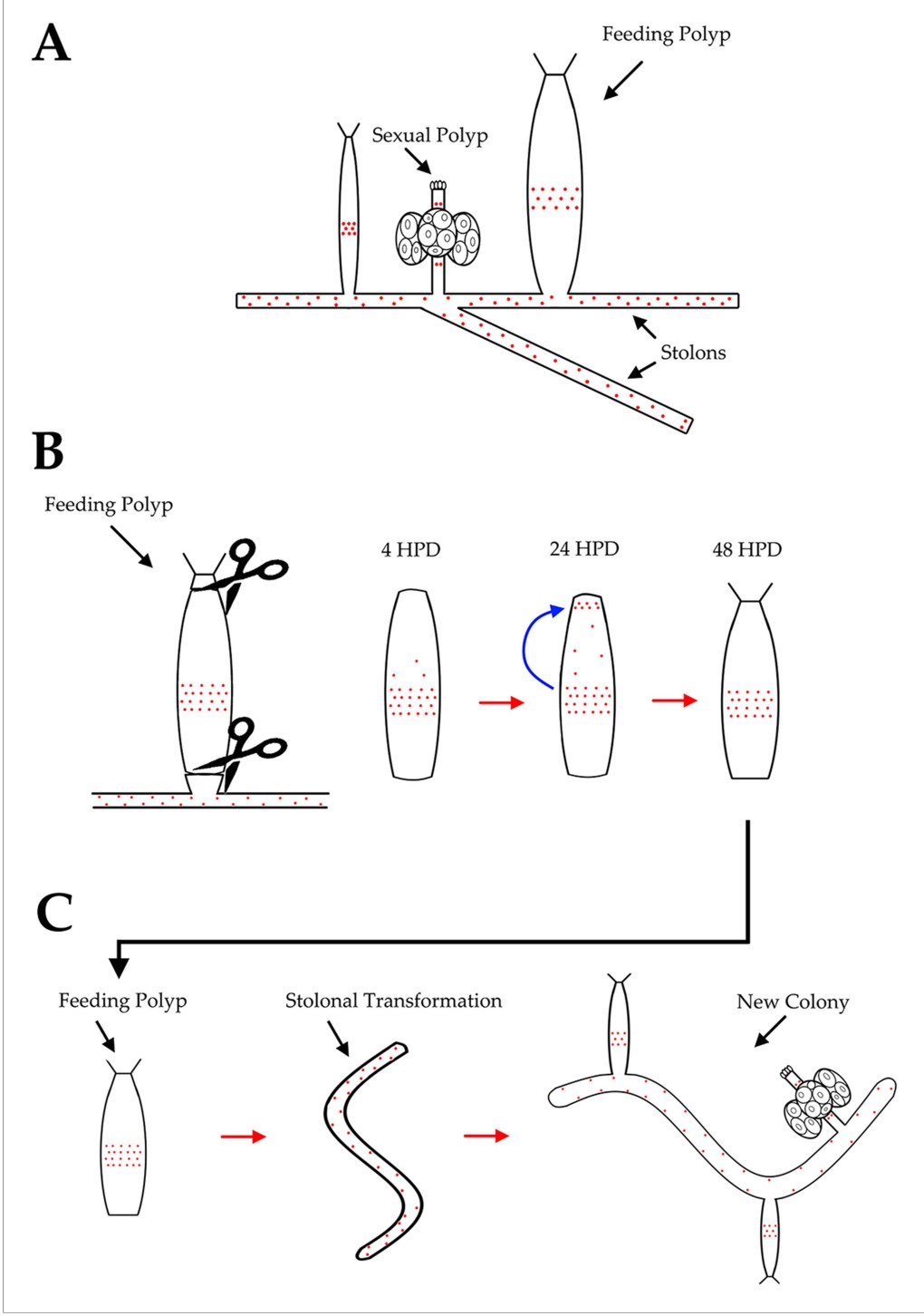

**Figure 7**. A summary of *Hydractinia* regeneration. Red dots represent proliferating i-cells. (**A**) A schematic of a normal colony including stolons, feeding and sexual polyps. (**B**) Head regeneration. Isolation of a polyp from the colony and its decapitation result in migration of i-cells to the head stump but not to the stolon stump. A head blastema, but not stolon blastema, is formed and provides the progenitors for the new head. (**C**) Transformation of polyps to stolons involves loss of polarity and ubiquitous spread of i-cells.

**Table 1.** Oligonucleotides used in this study

| Target | Accession | Primer name | Primer sequence |
|---|---|---|---|
| Piwi1 | JG772275.1 | CniwiForT7 | 5′-gatcataatacgactcactatagggagagttgatttcacaatcggttagac-3′ |
| | | CniwiRevSp6 | 5′-agtgcatttaggtgacactatagaagtgtactactactactactggttattt-3′ |
| | | PiwiIRNAiT7Fw | 5′-gcgtaatacgactcactatagggagagctgtgtggaaagaccagtc-3′ |
| | | PiwiIRNAiSP6Rv | 5′-tgcatttaggtgacactatagaagtgcgtcaaatccaatcaccatc-3′ |
| | | Piwi1qPCRfw | 5′-aagtatggcctggcatctca-3′ |
| | | Piwi1qPCRrv | 5′-cactgtctgctgtcgtaaaacc-3′ |
| | | Piwi1qPCRprobe | 5′-tgcagtatgaacaagatgtgatgttgtgtgctgatgttcag-3′ |
| Pl10 | AB048849.1 | Pl10ForT7 | 5′-gatcataatacgactcactatagggagattctggcaaaacagctgcattt-3′ |
| | | Pl10RevSP6 | 5′-agtgcatttaggtgacactatagaagtgagcttcatccagacaaagaaac-3′ |
| | | Pl10rnaiFWT7 | 5′-gcgtaatacgactcactatagggagagcgtaacacccattttg-3′ |
| | | PL10rnaiRVsp6 | 5′-tgcatttaggtgacactatagaagtgtaaatcacgcgca-3′ |
| Ncol1 | JX486117.1 | Ncol1-T7fwd | 5′-gatcataatacgactcactatagggacgtccaggaccaccaggagta-3′ |
| | | Ncol1-Sp6rev | 5′-tagcaatttaggtgacactatagaactgggcaacagtattgtggacaaga-3′ |
| Vasa | EF467228.1 | HeVASAforT7 | 5′-taatacgactcactatagggagaaggttcaaagtggttgccattt-3′ |
| | | HeVASArevSP6 | 5′-atttaggtgacactatagaagagtactgccaactttaccaat-3′ |
| | | VasaRNAiFWT7 | 5′-gcgtaatacgactcactatagggagagttgaaatgctgggacaagaagg-3′ |
| | | VasaRNAiRVsp6 | 5′-tgcatttaggtgacactatagaagtggcggtagcgataagaacagtc-3′ |
| Wnt3 | AM279678.1 | Wnt3ForwardPrimerT7 | 5′-gatcataatacgactcactataggggagtccgccttcattagtgg-3′ |
| | | Wnt3ReversePrimerSP6 | 5′-tagcaatttaggtgacactatagaatgggcggagtcgtatctatc-3′ |
| cMyc | JF820068.1 | cMycInSituFWt7 | 5′-gatcataatacgactcactatagggcctttaacgcctcccagttct-3′ |
| H2A | | H2aRNAiFwd1T7 | 5′-gatcataatacgactcactatagggatgtctggacgtggaaaagg-3′ |
| | | H2aRNAiRv1SP6 | 5′-tagcaatttaggtgacactatagaaccaatatctcagcagataaatattccaag-3′ |
| | | H2aRNAiFwd2T7 | 5′-gatcataatacgactcactatagggggagttggctggtaacgcag-3′ |
| | | H2aRNAiRv2SP6 | 5′-tagcaatttaggtgacactatagaaacttcttctgtcctttgtcgttct-3′ |
| H4 | | H4RNAiFwd1T7 | 5′-gatcataatacgactcactatagggatgtctggacgcggaaaag-3′ |
| | | H4RNAiRv1SP6 | 5′-tagcaatttaggtgacactatagaactttagtacacctctggtttcctc-3′ |
| | | H4RNAiFwd2T7 | 5′-gatcataatacgactcactataggggtcaaacgtatctctggccttat-3′ |
| | | H4RNAiRv2SP6 | 5′-tagcaatttaggtgacactatagaaaacctccgaatccgtaaagag-3′ |
| | | cMycInsituRVsp6 | 5′-agtgcatttaggtgacactatagaattgttaacggaaaagggaaaactg-3′ |
| Plasmid | | gfpRv-SAC1 | 5′-aaaaagagctcctatttgtatagttcatccatgccatg-3′ |
| | | TerminatorFw-PacI | 5′-aaaaattaattaacgtacgggcccttttcgtct-3′ |
| Race | | NewsplicedleaderFwd | 5′-tactcacactatttctaagtccctgagtttaag-3′ |
| | | PiwiRV2SP6 | 5′-tagcaatttaggtgacactatagaaccttagcgccacctgtgc-3′ |
| Cloning | | LigDVectorGFP-Fusion | 5′-gcggccgctgcagccccggt-3′ |
| | | BackbonelactRV1 | 5′-actggccgtcgttttacaac-3′ |
| | | Piwi1ProFw1new | 5′-cagatgatccgcagacaatagac-3′ |
| | | Piwipromrevin | 5′-gtttttcttcttataattttttctaaaaactt-3′ |
| | | PiwiRv2SP6 | 5′-tagcaatttaggtgacactatagaaccttagcgccacctgtgc-3′ |
| | | Piwi1TerRV1-PacI | 5′-aaaaattaattaagaaggcttacgctagtgtgaattag-3′ |
| | | Piwi1TerFw1-SAC1 | 5′-aaaaagagctcgtagctgcgcgttgtttacg-3′ |
| | | GFPSeqFusRev | 5′-ttgcatcaccttcaccctctcc-3′ |
| | | PBIGFor | 5′-taaaaataggcgtatcacgaggccc-3′ |

## TUNEL

TUNEL staining was complete as per the manufacture's protocol (Life Technologies). Click-iT TUNEL Alexa Fluor 488 Imaging Assay, Cat: 10245.

## Immunohistochemistry

Animals were fixed in 4% paraformaldehyde in PBS for 60 min and then washed three times in phosphate buffered saline - 0.3% Triton (PBST) and blocked for 30 min in 2% BSA/PBST. Primary antibodies (anti-Hiwi [a kind gift from Dr Celina Juliano, Yale University], anti acetylated tubulin [T7451, Sigma], anti-phospho H3 [ab5176 Abcam]) were diluted 1:500–1:2000 in BSA/PBST and incubated overnight at 4°C, followed by three washes with PBST then blocked for 30 min in 5% serum in BSA/PBST. Secondary antibodies (Alexa Fluor 488 goat anti-rabbit IgG [A-11008, Invitrogen], Alexa Fluor 546 Phalloidin LifeTec [A22283]) were diluted 1:500 in BSA/PBST/serum and incubated for 1 hr at room temperature. Animals were washed three times with PBST, incubated in 1:2000 Hoescht (20 mg/ml), DAPI (1:5000) or phalloidin (1:2000) in PBST and washed a further three times in PBST. Animals were mounted in mounting medium (F4680, Sigma).

## In situ hybridization

In situ hybridization was performed as previously described (*Gajewski et al., 1996*). Templates for DIG-labeled RNA probes (Roche) were generated by PCR. RNA synthesis was performed by SP6 and T7 RNA polymerases according to the manufacturer's protocol (Fermentas). The sequence of the oligonucleotides is given in *Table 1*. In situ hybridization was performed at 55°C.

## In vivo microscopy

Animals were embedded in Ibidi (Martinsried, Germany) μ-dish (#81156) using 1% low-melt agarose in seawater. They were observed using either a fluorescence stereomicroscope, DeltaVision deconvolution microscope, or Andor spinning disc confocal microscope. For time-lapse videos, images or stacks were taken every 5 min.

## RNAi

RNAi was performed as previously described (*Duffy et al., 2010*; *Millane et al., 2011*; *Duffy, 2012*). Templates for RNA synthesis were generated by PCR (see oligonucleotide list on *Table 1*). Sense and antisense RNA strands were generated as for in situ hybridization but were annealed by heating them together to 70°C and allowing them to cool down at room temperature. Animals were incubated in seawater to which dsRNA at 20–40 μg/ml was added directly after decapitation. The experiments run until the control animals had regenerated (usually between 3–5 days). dsRNA solution was replaced every 24 hr.

## Quantitative, real-time PCR

mRNA was extracted using standard Trizol/chloroform extraction technique and cleaned over RNeasy minikit (74104; Qiagen) according to the manufacturer's protocol. RNA was reverse transcribed using Omniscript RT kit (205110; Qiagen). qPCR was run on a StepOne Plus (Life Technologies) using TaqMan chemistry. Experiments were performed on three colonies using three technical replicates for each.

## Generating transgenic animals

We cloned the genomic regions 2.5 kb upstream and 1.1 kb downstream of the *Hydractinia Piwi1* coding sequence into a modified pBluescript backbone (*Künzel et al., 2010*) and replaced the coding sequence by *GFP*. (*Figure 5—figure supplement 2*). One-cell stage embryos were microinjected with 200 pl volume of the plasmid at a concentration of 4–5 μg/μl as previously described (*Künzel et al., 2010*; *Millane et al., 2011*; *Duffy, 2012*; *Kanska and Frank, 2013*).

## Acknowledgements

We thank members of our lab for assistance and discussions. Anti-hydra Piwi (Hywi) antibodies were a kind gift from Celina Juliano, Yale University. Elaine Dunleavy (Centre for Chromosome Biology, NUIG) is kindly acknowledged for assistance with DeltaVision microscopy.

# Additional information

### Funding

| Funder | Grant reference | Author |
|--------|-----------------|--------|
| Science Foundation Ireland (SFI) | 11/PI/1020 | Uri Frank |
| Irish Higher Education Authority | Programme for Research in Third Level Institutions | Kerry Thompson |

The funders had no role in study design, data collection and interpretation, or the decision to submit the work for publication.

### Author contributions

BB, UF, Conception and design, Acquisition of data, Analysis and interpretation of data, Drafting or revising the article; KT, Acquisition of data

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
