## [Decision Letter]

Thank you for sending your work entitled “Distinct mechanisms underlie oral versus aboral regeneration in the cnidarian *Hydractinia echinata*” for consideration at *eLife*. Your article has been favorably evaluated by Janet Rossant (Senior editor), Alejandro Sánchez Alvarado (Reviewing editor), and two reviewers.

The Reviewing editor and the reviewers discussed their comments before we reached this decision, and the Reviewing editor has assembled the following comments to help you prepare a revised submission.

This manuscript characterizes the stem cells for regeneration in the cnidarian model *Hydractinia echinata*. This is perhaps the first time a detailed study of regenerative processes has been carried out in this marine colonial hydrozoan polyp. The authors report on the cellular and molecular processes underlying two distinct regenerative processes that take place in the same animal: 1) oral regeneration after decapitation of the head of the polyps; and 2) aboral regeneration when the polyps are detached at their aboral pole from the colony. An important aspect of this work is the establishment that *Piwi*-expressing i-cells migrate from the i-cell belt to the anterior amputation plane to participate in regeneration. The authors pioneer live imaging of the i-cells in *Piwi*:GFP transgenic reporters to establish this phenomenon. They also use irradiation to establish that proliferative cells are required for regeneration in *Hydractinia*. In addition they use RNAi of several stem cell associated factors and show that while EdU incorporation does not seem to be affected, regeneration is suppressed. This work is a major step in the dissection of molecular processes underlying regeneration in cnidarians. As such, we would like the authors to fully address the following comments:

1) The phenotypes shown by the authors are quite convincing but the results shown in Figure 2 should be supported by quantitative data and statistical analysis regarding the different morphological and cellular phenotypes and kinetics of regeneration are missing. Similarly results shown in Figure 3 could be more precise, as deduced from the combination of EDU pulse chase labeling to PH3 detection that could provide a temporal analysis of the contribution of cycling cells in blastema formation. In fact the pattern of cycling cells on detached but intact polyps appears variable, in some cases forming a clear and unique band relatively low along the body column with sharp boundaries (as in Figure 3), while in other cases, it appears rather spread all along the body column of the polyps as shown in Figure 3—figure supplement 1 (panels B and C). As this band pattern is quite important in the blastema formation mechanism proposed by the authors, this variability should be quantified and discussed by the authors. Chiefly, we would like to see a more rigorous quantification of the cell cycle parameters measured during regeneration.

2) It seems important to the arguments made by the authors to be able to monitor the interstitial to epithelial transition in the blastema. We would like to suggest that the authors monitor the appearance of epithelial markers in cells that are still GFP positive at the protein level but no longer at the transcript level. That way the authors could show that i-cells from the blastema differentiate into epithelial cells when the *Piwi* promoter becomes inactive. To get this result, the GFP protein should persist at least one or two days after transcripts disappear, that way the *Piwi*:GFP(+) i-cells would lose their interstitial GFP signature at the time they gain the epithelial one.

3) Some specificity controls for the RNAi experiments need to be presented. Since RNAi of the stem cell factors all have the same phenotype, it is suggested to knockdown a key cell cycle regulator and show that this affects EdU. It is also suggested to knockdown a gene that is not expressed in the i-cells and show that it has a different phenotype.

---

## [Author Response]

*1) The phenotypes shown by the authors are quite convincing but the results shown in*
Figure 2
*should be supported by quantitative data and statistical analysis regarding the different morphological and cellular phenotypes and kinetics of regeneration are missing. Similarly results shown in*
Figure 3
*could be more precise, as deduced from the combination of EDU pulse chase labeling to PH3 detection that could provide a temporal analysis of the contribution of cycling cells in blastema formation. In fact the pattern of cycling cells on detached but intact polyps appears variable, in some cases forming a clear and unique band relatively low along the body column with sharp boundaries (as in*
Figure 3*), while in other cases, it appears rather spread all along the body column of the polyps as shown in*
Figure 3—figure supplement 1
*(panels B and C). As this band pattern is quite important in the blastema formation mechanism proposed by the authors, this variability should be quantified and discussed by the authors. Chiefly, we would like to see a more rigorous quantification of the cell cycle parameters measured during regeneration*.

Regeneration experiments as described in Figure 2 have been carried out in our lab hundreds of times over the past years. These experiments have established that over 90% of healthy *Hydractinia* polyps regenerate their heads following decapitation within two-three days, but regeneration times in unhealthy or malnourished animals may last substantially longer. By contrast, very young polyps (i.e. shortly post metamorphosis) may regenerate an amputated head even faster, perhaps because at this stage they contain many more i-cells than fully grown ones. We have added some clarification statements to the text to address this point (in the first Results section).

As suggested, we performed anti-pH3 analysis of EdU pulse chased cells during head regeneration. Our results, presented in a new supplemental figure to Figure 5 (Figure 5—figure supplement 1), show that the blastema includes double positive cells, i.e. cells that were in S-phase in the band before decapitation, and in mitosis in the head blastema 24 and 48 hours later. These cells had migrated from the i-cell band to the blastema within this time frame.

The distribution of cycling cells in *Hydractinia* polyps largely depends on developmental stage and state of health. In young, post metamorphosed polyps one can find cycling cells throughout the body column and tentacles although the majority is still found in an aboral band. This is also true for newly growing, clonal polyps in a sexually mature colony. However, fully-grown polyps generally display a clear proliferation ‘band’ at the lower polyp body column, co-localizing with cells expressing stem cells associated genes. The width of the ‘band’ is variable, but cycling cells are only rarely observed in the head area or tentacles. As requested, we have now quantified it in an additional experiment and found that over 90% of all randomly selected polyps had displayed this pattern of proliferating cells. This is now also mentioned in the text.

*2) It seems important to the arguments made by the authors to be able to monitor the interstitial to epithelial transition in the blastema. We would like to suggest that the authors monitor the appearance of epithelial markers in cells that are still GFP positive at the protein level but no longer at the transcript level. That way the authors could show that i-cells from the blastema differentiate into epithelial cells when the* Piwi *promoter becomes inactive. To get this result, the GFP protein should persist at least one or two days after transcripts disappear, that way the* Piwi*:GFP(+) i-cells would lose their interstitial GFP signature at the time they gain the epithelial one*.

Evidence for interstitial to epithelial transition in *Hydractinia* has been provided in at least two publications in the past ([36], Dev Biol 275, 2015-224; [27], Dev Biol 348, 120-129). Therefore, demonstrating that *Piwi1*+ i-cells can give rise to epithelial cells would not constitute a major scientific novelty in our opinion. In the regenerative context, however, we agree that we cannot exclude that epithelial cell proliferation and/or transdifferentiation had also contributed to the new head. Unfortunately, in the absence of a verified, exclusive marker for *Hydractinia* epithelial cells we are, at present, unable to perform the suggested experiments. Furthermore, we feel that the question of possible contribution of differentiated epithelial cells to head regeneration is not central to the point we are making in our paper. We have therefore toned-down our statement on the role of epithelial cells in head regeneration in the manuscript and hope to address this question in a future study.

*3) Some specificity controls for the RNAi experiments need to be presented. Since RNAi of the stem cell factors all have the same phenotype, it is suggested to knockdown a key cell cycle regulator and show that this affects EdU. It is also suggested to knockdown a gene that is not expressed in the i-cells and show that it has a different phenotype*.

To strengthen our claim of RNAi specificity we used two new approaches in addition to the non-coding dsRNA (control RNAi) that has already been presented as control at first submission.

First, we performed qPCR on *Piwi1* RNAi animals to confirm a specific reduction of *Piwi1* transcript level (normalized to 18S). Second, as suggested, we knocked down two additional genes we expected to be important for cell cycle progression: histone H2A and histone H4. Indeed, treatment with H2A or H4 dsRNA almost completely abolished EdU incorporation comparing to non-coding dsRNA we used as control. These new results are now presented in a new supplemental figure to Figure 4 (Figure 4—figure supplement 1).

Knocking down of any confirmed non-i-cell gene during regeneration might very well result in compromised regeneration. In particular, *Ncol1* (an exclusive nematoblast marker) knockdown did affect regeneration as we have shown and this would probably also be true for neural genes because *Hydra* head regeneration requires de novo neurogenesis ([32], Development 134, 1191-1201). Hence, it would be difficult to *a priori* select a gene that does not have a role in head regeneration since this complex process probably involves many genes.

We would also like to add that in the past we have used RNAi to knockdown various *Hydractinia* genes in various contexts and monitored distinct phenotypes for each knocked down gene (e.g. [16], Development 137, 3057-3066; [15], Dev Biol 362, 271-281; [33], Development 138, 2419-2439).

We therefore suggest that the new experiments described above, together with our published work, provide strong evidence for the specificity of RNAi in *Hydractinia*.